# Realization of unpinned two-dimensional dirac states in antimony atomic layers

Qiangsheng Lu[1], Jacob Cook[1], Xiaoqian Zhang[1], Kyle Y. Chen[2], Matthew Snyder[1], Duy Tung Nguyen[1], P. V. Sreenivasa Reddy [3], Bingchao Qin [4], Shaoping Zhan[4], Li-Dong Zhao [4], Pawel J. Kowalczyk [5] ✉, Simon A. Brown [6], Tai-Chang Chiang [7,8], Shengyuan A. Yang [9], Tay-Rong Chang[3] & Guang Bian [1] ✉

Two-dimensional (2D) Dirac states with linear dispersion have been observed in graphene and on the surface of topological insulators. 2D Dirac states discovered so far are exclusively pinned at high-symmetry points of the Brillouin zone, for example, surface Dirac states at $\bar{\Gamma}$ in topological insulators $Bi_2Se(Te)_3$ and Dirac cones at $K$ and $K'$ points in graphene. The low-energy dispersion of those Dirac states are isotropic due to the constraints of crystal symmetries. In this work, we report the observation of novel 2D Dirac states in antimony atomic layers with phosphorene structure. The Dirac states in the antimony films are located at generic momentum points. This unpinned nature enables versatile ways such as lattice strains to control the locations of the Dirac points in momentum space. In addition, dispersions around the unpinned Dirac points are highly anisotropic due to the reduced symmetry of generic momentum points. The exotic properties of unpinned Dirac states make antimony atomic layers a new type of 2D Dirac semimetals that are distinct from graphene.

Two-dimensional (2D) Dirac states have attracted tremendous research interests since the discovery of graphene and topological insulators[1–5]. The linear dispersion and vanishing effective mass of Dirac fermion states are connected to interesting physical phenomena[4]. To date, gapless 2D Dirac states have only been observed in graphene[1,2] and on the surface of topological insulators[3,6]. The Dirac states in those 2D systems are exclusively pinned at high-symmetry points of the Brillouin zone, such as $K(K')$ of graphene (due to the $C3$ rotational symmetry). The Dirac states also feature isotropic low-energy dispersion due to the local rotational symmetry of the crystal lattice. The warping effect, a higher-order correction to the linear

dispersion, becomes prominent only for states distant from the Dirac point[7]. These features impose constraints on applications of the massless Dirac states. For example, the two Dirac cones in graphene are pinned at opposite corners of the Brillouin zone. It is difficult to make two valleys effectively coupled in monolayer graphene. Therefore, 2D Dirac states that are unpinned in momentum space are desirable for enabling novel functionalities in Dirac materials.

Recently, it has been theoretically proposed that multiple 2D Dirac states emerge at generic momentum points in the low-energy spectrum of group-Va few-layers with phosphorene-like lattice structure[8]. In this case, highly anisotropic cone is a character of the

[1]Department of Physics and Astronomy, University of Missouri, Columbia, MO 65211, USA. [2]Rock Bridge High School, Columbia, MO 65203, USA. [3]Department of Physics, National Cheng Kung University, Tainan 701, Taiwan. [4]School of Materials Science and Engineering, Beihang University, Beijing 100191, China. [5]Department of Solid State Physics, Faculty of Physics and Applied Informatics, University of Lodz, 90-236 Lodz, Pomorska 149/153, Poland. [6]The MacDiarmid Institute for Advanced Materials and Nanotechnology, School of Physical and Chemical Sciences, University of Canterbury, Private Bag 4800, Christchurch 8140, New Zealand. [7]Department of Physics, University of Illinois at Urbana-Champaign, Urbana, IL 61801-3080, USA. [8]Frederick Seitz Materials Research Laboratory, University of Illinois at Urbana-Champaign, 104 South Goodwin Avenue, Urbana, IL 61801-2902, USA. [9]Research Laboratory for Quantum Materials, Singapore University of Technology and Design, Singapore 487372, Singapore. ✉e-mail: pawel.kowalczyk@uni.lodz.pl; biang@missouri.edu

unpinned Dirac point, because the generic $k$ point has much reduced local symmetry. The unpinned nature makes the Dirac nodes movable in momentum space, e.g., by lattice strains. All these properties lead to tunable transport properties of Dirac states, which are unavailable in conventional 2D Dirac materials. Therefore, the Dirac states in group-Va few-layers are distinct from those in graphene and thus, offer new insights into the Dirac fermion physics at low dimensions. In this work, we report the observation of multiple unpinned Dirac states near the Fermi level in single layer (1L) and double layer (2L) antimony (Sb) films in the phosphorene structural phase.

A group-Va pnictogen atom typically forms three covalent bonds with its neighbors. In the 2D limit, two allotropic structural phases, the orthorhombic phosphorene-like phase[9,10] and the hexagonal honeycomb-like phase[11,12], are allowed by this requirement. The two phases are referred to as $\alpha$ and $\beta$-phases, respectively, in the literature. In the family of group-Va elements, P, As, Sb and Bi can form phosphorene-like structures[9,10,13–19]. Here we focus on the phosphorene-like $\alpha$-antimonene ($\alpha$-Sb for short). As shown in Fig. 1a, in the lattice of 1L $\alpha$-Sb, the three Sb-Sb bonds form a tetrahedral configuration. This results in two atomic sublayers with a vertical separation comparable to the inplane bond length. In each atomic sublayer, the bonding between Sb atoms forms zig-zag chains along the $y$-direction. The unit cell, marked by the blue rectangle in Fig. 1a, has a four-atom basis (two in each atomic plane). The space group of the lattice is $D_{2h}(7)$, which includes space inversion $P$, a vertical mirror plane $M_y$ perpendicular to $\hat{y}$, two two-fold rotational axes $C_{2y}$ and $C_{2z}$, and a glide mirror $\tilde{M}_z$ that is parallel to the $x$–$y$ plane and lies in the middle between the two atomic planes. The glide mirror reflection is composed of a mirror reflection and an in-plane translation by ($0.5a$, $0.5b$)[20], and it interchanges the two atoms connected by the red arrow in Fig. 1a. Previous works[20,21] have shown that the glide mirror of bismuthene and antimonene leads to band crossings at certain high-symmetry momentum points of the Brillouin zone. However, those nodal points are >0.5 eV away from the Fermi level and thus irrelevant to the transport properties of the materials. In this work, we demonstrate by angle-resolved photoemission spectroscopy (ARPES) the existence of unpinned Dirac states at the Fermi level in $\alpha$-Sb films, which makes this materials an ideal 2D Dirac semimetal. The Dirac

states are located at generic momentum points. The location of Dirac points can be shifted by lattice strains, which is confirmed by our ARPES experiments.

## Results

We grew $\alpha$-Sb thin films by the method by molecular beam epitaxy (MBE). $\alpha$-Sb has been grown on various substrates[22–24]. In this work, we chose SnS as substrate, because SnS(001) surface has similar lattice parameters and lattice symmetry as $\alpha$-Sb, and thus, favors the formation of $\alpha$-Sb. The (001) surface lattice constants of SnS are $a = 4.35$ Å and $b = 3.99$ Å[25]. The crystallographic structure of the epitaxial sample is shown in Fig. 1a. The in-plane lattice constants of 1L $\alpha$-Sb are $a = 4.42$ Å and $b = 4.30$ Å in the $x$ and $y$-directions, respectively, while the in-plane nearest-neighbor bond length is 2.85 Å. Figure 1b, c show the STM image of two $\alpha$-Sb/SnS samples. The first sample consists of mainly 1L $\alpha$-Sb islands, as shown in Fig. 1c. The height profile taken along the red arrow (shown in Fig. 1d) shows that the apparent height of the 1L $\alpha$-Sb island on the SnS surface is 6.1 Å. The surface unit cell of SnS substrate can be seen in the zoom-in STM image shown in Fig. 1e. The second sample possesses both 1L and 2L domains as shown in Fig. 1c. The atom-resolved images taken from the 1L and 2L domains (Fig. 1f, g) clearly demonstrate the rectangular surface unit cell of $\alpha$-Sb. The height profile taken along the green arrow in Fig. 1c shows the apparent height of the second layer is 6.8 Å, which is slightly larger than that of the first layer due to the lattice relaxation.

To study the electronic band structure, we performed first-principles calculations for the band structure of 1L and 2L $\alpha$-Sb films. The calculated band structure of 1L $\alpha$-Sb is shown in Fig. 2. In the absence of SOC, the bottom conduction band and the top valence band at $\bar{\Gamma}$ are separated by an energy gap of 0.3 eV, as shown in Fig. 2a. The two bands are dominated of the $p_z$-orbital character near the zone center[8]. Between $\bar{\Gamma}$ and $\bar{X}_1$, there is a small hole pocket at the Fermi level, which is generated by the overlap of a pair of bands of mainly the $p_{x,y}$-orbital character[8]. To see this pocket more clearly, we plot the zoom-in band structure along the lines of 'cut1' and 'cut2' marked in Fig. 2m. The conduction and valence bands cross each other and leave a pair of nodal points, which lie 0.15 eV above the Fermi level, as shown in Fig. 2b, c. It is worth noting that the Dirac

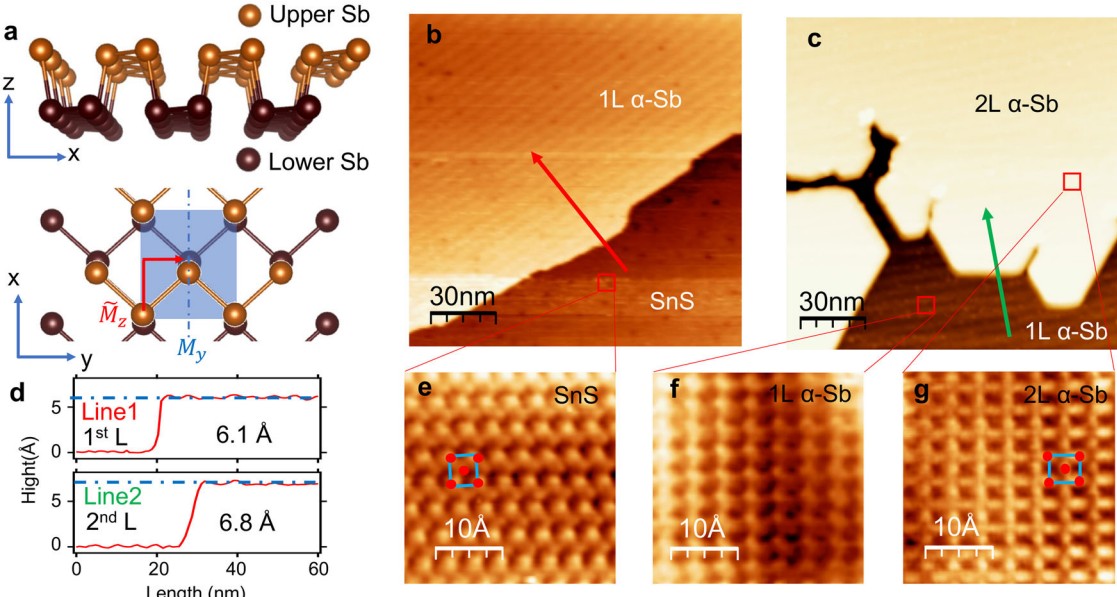

**Fig. 1 | Lattice structure and STM characterization of $\alpha$-Sb films. a** Side and top views of $\alpha$-Sb lattice structure. **b** STM image of 1L $\alpha$-Sb grown on SnS substrate. **c** STM image of 1L and 2L $\alpha$-Sb domains. **d** The height profiles taken along the red arrow in **b** and the green arrow in **c**. **e–g** Atom-resolved STM images taken from SnS surface, 1L $\alpha$-Sb and 2L $\alpha$-Sb domains.

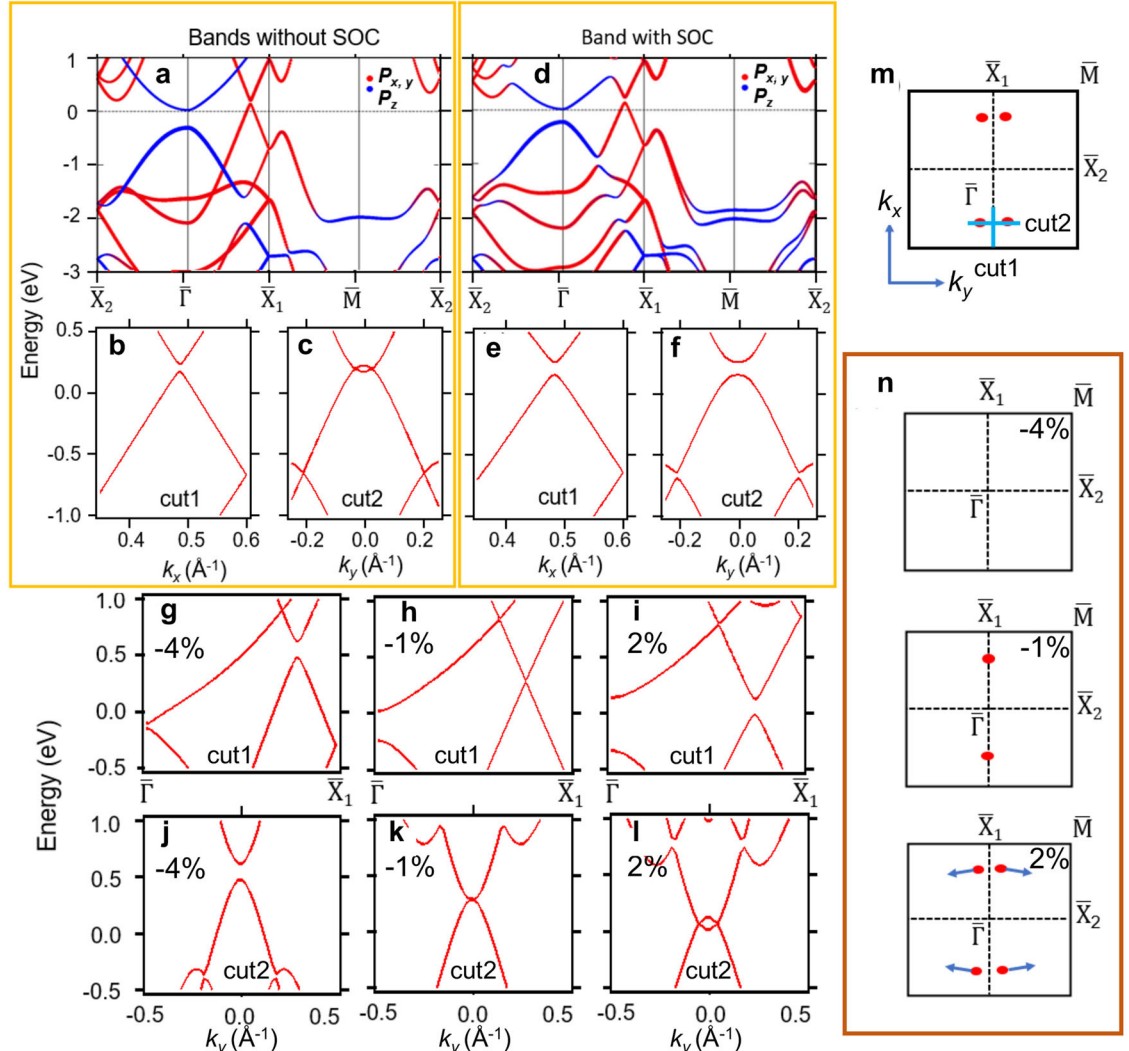

**Fig. 2 | Calculated band structure of 1L α-Sb. a** The band structure of 1L α-Sb calculated without the inclusion of SOC. The bands are colored according to their orbital components. **b**, **c** The calculated band dispersion along 'cut1' and 'cut2' (marked in **m**) without the inclusion of SOC. **d**–**f** Same as **a**–**c**, but calculated with the inclusion of SOC. **g**–**l** Band structure (without SOC) of 1L α-Sb under uniaxial lattice strains along y-direction. **g**, **j** Band dispersion along 'cut1' and 'cut2'

(marked in **m**) with −4% lattice strain in ŷ direction. **h**, **k** and **i**, **l** same as **g**, **j**, but for lattice strains in ŷ direction of −1% and +2%, respectively. **m** The Brillouin zone of 1L α-Sb. The Dirac points in the case without SOC are marked by the red dots. **n** The locations of Dirac points under uniaxial lattice strains of −4%,−1% and +2% in y-direction. The blue arrows indicate the movement of Dirac points as the lattice constant b increases.

nodes are located at generic momentum points, as schematically shown in Fig. 2m. In the absence of SOC, the Dirac nodes are stable due to the protection by the spacetime inversion symmetry $PT$, which enforces a quantized Berry phase $\theta_B$ for each Dirac point. The Berry phase along a closed loop $\ell$ encircling each Dirac node is defined as follows,

$$\theta_B = \oint_\ell \boldsymbol{A_k} \cdot \mathrm{d}\boldsymbol{k} = \pm\pi, \tag{1}$$

where $A_k$ is the berry connection of the occupied valence bands[8]. On the other hand, with the inclusion of SOC, the number of bands is doubled due to the spin degrees of freedom. Gap-opening terms are allowed in the Hamiltonian to lift the band degeneracy at Dirac points. This can be seen in the band structure plotted in Fig. 2d–f. Though the hole pockets remain largely unchanged at the Fermi level, the conduction and valence bands are separated by an SOC-induced energy gap of 70 meV. The gapped band dispersion is similar to the quasi-2D Dirac states observed in bulk crystal ZrSiS[26]. Below the Fermi level, the lower part of the gapped Dirac cone remains nearly linear.

The band dispersion close to the Dirac nodes in absence of SOC can be described by the linear Hamiltonian,

$$\widetilde{H}(\boldsymbol{k}) = v_x k_x \sigma_y + \omega v_y (k_y - \omega k_0)\sigma_x, \tag{2}$$

where $k_x$ is measured from the location of the Dirac points, $\omega = \pm 1$ indicates the opposite chirality of the two Dirac nodes along 'cut2' in Fig. 2m, $k_0$ measures the separation between the two Dirac nodes in $k_y$ direction, $\sigma_{x,y}$ are the Pauli matrices for pseudospin, and $v_{x,y}$ is the group velocity at the Dirac point in $k_x$ and $k_y$ directions, respectively. According to the calculated band structure in Fig. 2a–c, $v_x = 8.35 \times 10^5$ m/s and $v_y = 4.08 \times 10^5$ m/s, indicating a highly anisotropic Dirac cone. To further examine the unpinned nature of the Dirac bands, we calculated the band structure under uniaxial lattice strains. The lattice constant b in ŷ direction is changed by −4%, −1% and +2%, where the '+' and '−' signs correspond to lattice expansion and compression, respectively. The results are plotted in Fig. 2g–l. Under the lattice stain of −4%, the conduction and valence bands are separated in energy, and thus no Dirac nodes are formed as shown in Fig. 2g, j. At the critical lattice strain of −1%, the conduction and valence bands

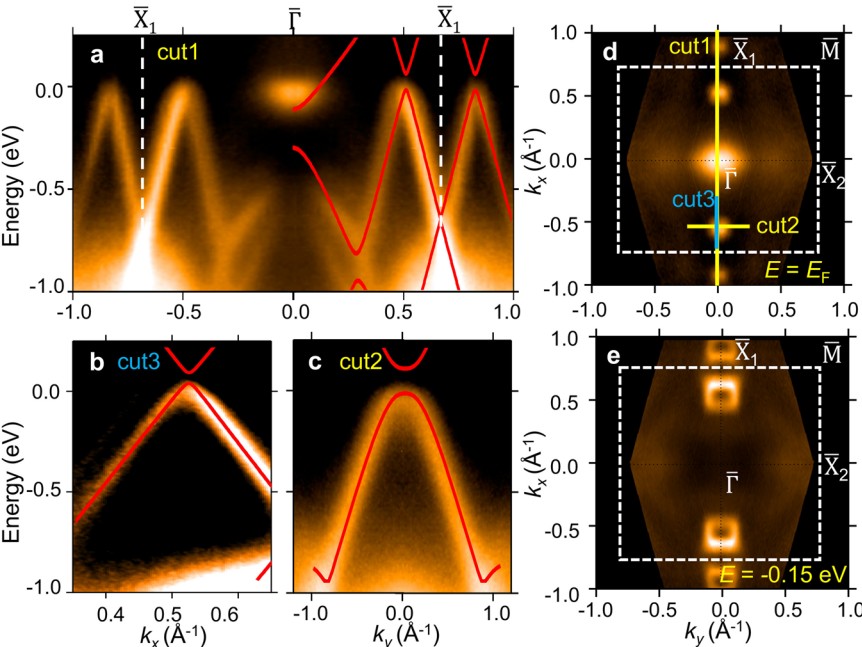

**Fig. 3 | ARPES spectrum of 1L $\alpha$-Sb. a** ARPES spectrum taken along $\bar{X}_1$-$\bar{\Gamma}$-$\bar{X}_1$. **b, c** Zoom-in spectra taken along 'cut3' and 'cut2' (marked in **d**), respectively. **d** Fermi surface taken from the 1L $\alpha$-Sb sample by ARPES. **e** ARPES iso-energy contour taken at $E = -0.15$ eV.

touch each other, leading to a quadratic Dirac cone (Fig. 2h, k), in which the band dispersion is linear in $k_x$ direction and quadratic in $k_y$ direction. The quadratic Dirac cone signals a topological transition between a semimetallic phase and a band insulator[27]. The quadratic band dispersion occurs when two Dirac points merge in a two-dimensional crystal. This can be seen from the fact that increasing the lattice constant $b$ from the critical value makes the quadratic Dirac node split into two linear Dirac nodes, as demonstrated in Fig. 2i, l. The separation between the two Dirac nodes is sensitive to the magnitude of lattice strains. The arrows in Fig. 2n indicates the movement of Dirac nodes when the lattice is further expanded in $\hat{y}$ direction. These results reveal two prominent features of unpinned Dirac states: (1) the Dirac nodes can freely move in the momentum space by perturbations such as lattice strains; and (2) the dispersions of Dirac bands are intrinsically highly anisotropic due to the reduced symmetry at the location of the generic Dirac points.

The ARPES result taken from the 1L $\alpha$-Sb sample is shown in Fig. 3. The photon energy is 21.2 eV. There are three prominent features on the Fermi surface, namely, one electron pocket at $\bar{\Gamma}$ and two hole pockets near $\bar{X}_1$ (Fig. 3d). The band spectrum taken along the $\bar{X}_1$-$\bar{\Gamma}$-$\bar{X}_1$ direction is plotted in Fig. 3a. The calculated band dispersion is overlaid on the ARPES spectrum for comparison. The theoretical bands agree with the ARPES spectrum, especially in showing the linear bands between $\bar{\Gamma}$ and $\bar{X}_1$. The molecular beam epitaxy (MBE) sample is slightly electron-doped due to the charge transfer from the SnS substrate to the $\alpha$-Sb overlayer, and thus the Fermi level of the calculated bands was shifted to match the ARPES spectrum. We note that the band calculation was performed with freestanding film geometry and experimental lattice constants. The good consistency between the calculated band structure and ARPES spectra indicates that the substrate coupling is weak, and thus, the epitaxial $\alpha$-Sb film can be considered as nearly freestanding. In the iso-energy contour taken at 0.1 eV below the Fermi level (Fig. 3e), there are only a pair of circular pockets sitting close to $\bar{X}_1$, which are the cross-sections of the lower Dirac cone. The zoom-in spectra along 'cut2' and 'cut3' are plotted in Fig. 3b, c. The ARPES spectra show clearly the linear band dispersion from the lower part of the gapped Dirac cone.

We also calculated the band structure of 2L $\alpha$-Sb. The results are summarized in Fig. 4. In the absence of SOC (Fig. 4a), there are multiple band crossings along the $\bar{\Gamma}$-$\bar{X}_1$ direction near the Fermi level. This is because the number of $p_{x,y}$-orbital dominated bands is doubled in the 2L film. The location of Dirac points is schematically plotted in Fig. 4k. The zoom-in bands along 'cut3' and 'cut4' in Fig. 4d, e clearly show the Dirac states close to the line of $\bar{\Gamma}$-$\bar{X}_1$. The Dirac cone in Fig. 4d is highly anisotropic. The Dirac node is on the $\bar{\Gamma}$-$\bar{X}_1$ line and at 0.2 eV above the Fermi level. The two branches of the Dirac cone are nearly degenerate in the direction of 'cut3'. On the other hand, the bands along 'cut4' (Fig. 4e) show a pair of Dirac nodes at 0.4 eV above the Fermi level, which are generated by the small overlap between the conduction and valence bands. The dispersion is similar to the Dirac bands of 1L $\alpha$-Sb in the absence of SOC. Moreover, there is another band crossing along the $\bar{\Gamma}$-$\bar{X}_2$ direction as shown in Fig. 4a. The bands at this new nodal point are of $p_z$ orbital character. In Fig. 4b, c, the zoom-in bands along 'cut1' and 'cut2' (marked in Fig. 4k) show that this new Dirac point is located at a generic momentum point between $\bar{\Gamma}$ and $\bar{X}_2$, and sit at 0.2 eV below the Fermi level. The Dirac band is gapless in the absence of SOC[8]. The band dispersions calculated with SOC are plotted in Fig. 4f, g. Energy gaps are opened at all Dirac nodes marked in Fig. 4k. Interestingly, the SOC gap found at the Dirac points of the $p_z$ band (Fig. 4g, h) is very small, only 6 meV. It can be explained by the fact that the $p_z$ band is largely immune to the effects of SOC, since the first-order SOC matrix element of the $p_z$ orbital vanishes, i.e.,

$$\left\langle \Psi_{p_z} | \boldsymbol{L} \cdot \boldsymbol{S} | \Psi_{p_z} \right\rangle = \left\langle \Psi_{p_z} | L_+ S_- + L_- S_+ + L_z S_z | \Psi_{p_z} \right\rangle = 0, \qquad (3)$$

where $\Psi_{p_z}$ is the wavefunction of the $p_z$ orbital with $L_z = 0$. We note that the $p_z$ orbitals do not hybridize with the $p_{x,y}$ orbitals since they have opposite parities of $\hat{M}_z$. The SOC matrix element is nonzero for the $p_{x,y}$ bands due to the mixing of $p_x$ and $p_y$ orbitals as schematically shown in Fig. 4l. Therefore, energy gaps can only be opened in the $p_z$ band by higher-order SOC effects induced by the inter-site hoppings[28]. As a result, the SOC gaps are highly suppressed in the $p_z$ band compared to those gaps found in the $p_{x,y}$ bands (Fig. 4i, j). Therefore, the unpinned Dirac bands formed by the $p_z$ orbitals is nearly gapless even in the presence of SOC.

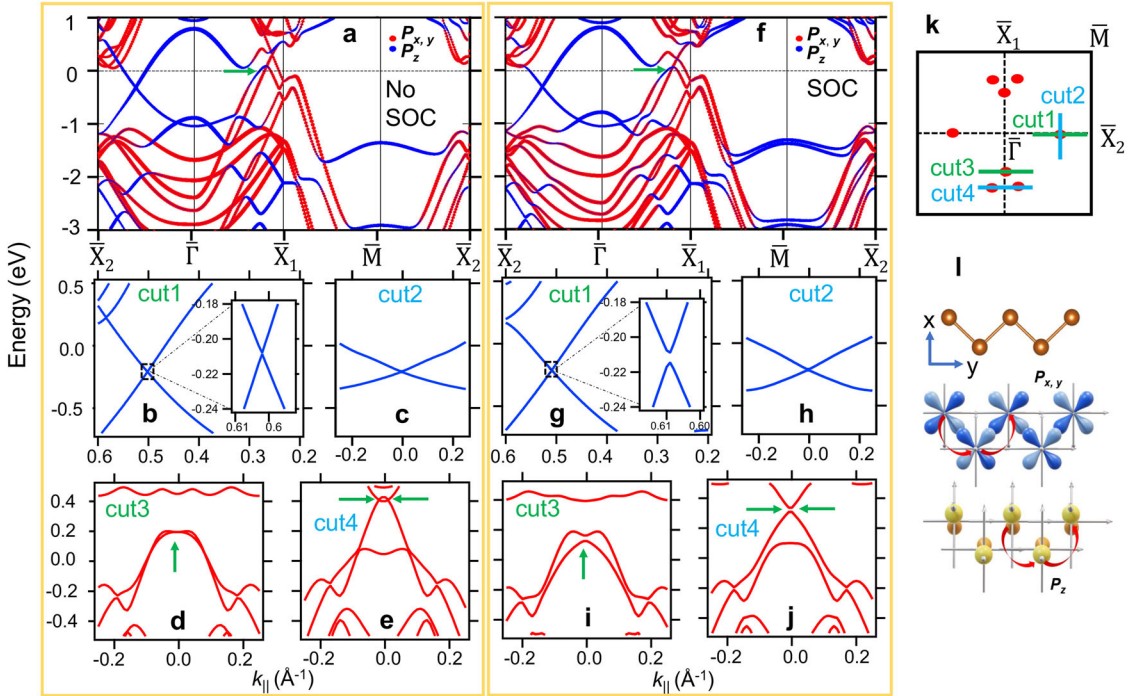

**Fig. 4 | Calculated band structure of 2L α-Sb. a** The band structure of 2L α-Sb calculated without the inclusion of SOC. **b–e** The calculated band dispersions along 'cut1-4', respectively. The Dirac points in the direction of $\bar{\Gamma}$-$\bar{X}_1$ are marked by green arrows in **d** and **e**. **f–j** Same as **a–e**, but calculated with the inclusion of SOC. **k** The Brillouin zone of 2L α-Sb. The Dirac points in the case without SOC are marked by red dots. **l** Schematic diagrams showing the formation of $p_{x,y}$ and $p_z$ bands.

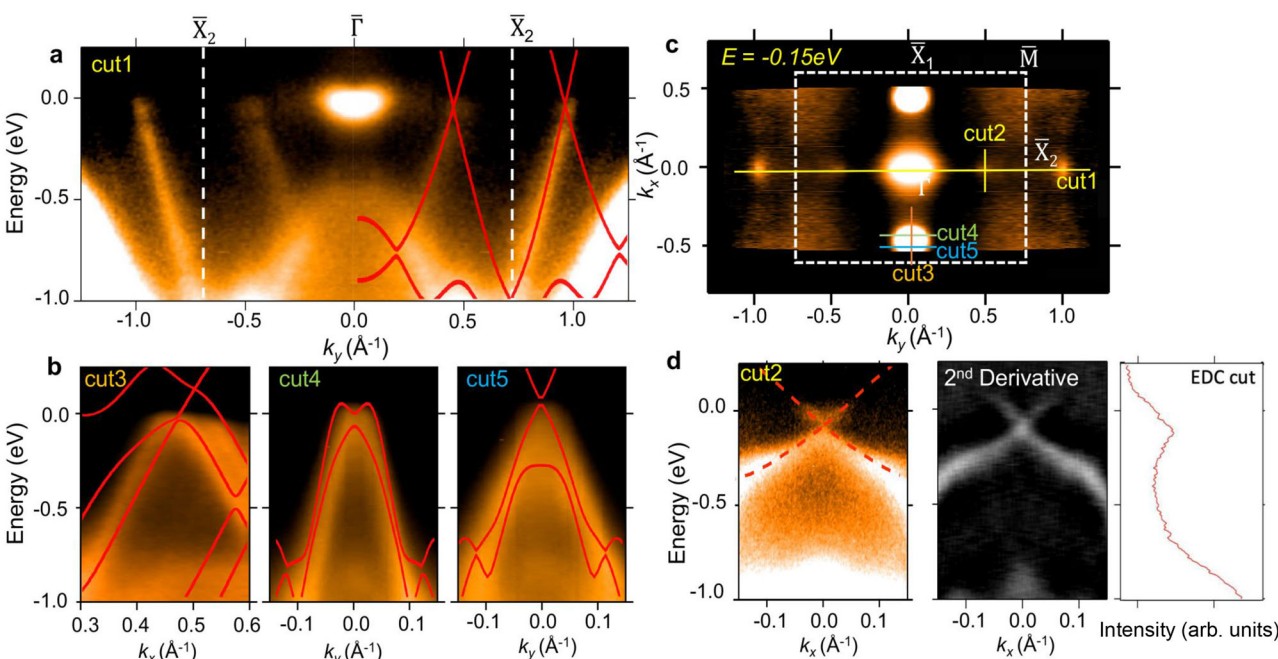

**Fig. 5 | ARPES spectrum of 2L α-Sb. a** ARPES spectrum taken along $\bar{X}_2$-$\bar{\Gamma}$-$\bar{X}_2$. **b** ARPES spectra taken along 'cut3-5' (marked in **c**). **c** ARPES iso-energy contour of 2L α-Sb at $E = -0.15$ eV. **d** ARPES spectrum taken along 'cut2' showing the Dirac bands of $p_z$ orbital character. The second derivative of the ARPES spectrum along 'cut2' is plotted for better visualization of the Dirac band structure. The energy distribution curve (EDC) taken at the momentum of the band crossing demonstrates the absence of a gap at the Dirac point.

We measured the band structure of the 2L α-Sb sample shown in Fig. 1c. The ARPES result is plotted in Fig. 5. The Fermi surface contour is plotted in Fig. 5c. We note that the sample consists of 1L and 2L domains. The bright pocket located at the zone center is from the 1L domains of the samples. The nodal features of 2L α-Sb at generic momentum points can also be easily identified. The band spectrum along the $\bar{X}_2$-$\bar{\Gamma}$-$\bar{X}_2$ direction (Fig. 5a) exhibits the linear Dirac band dispersion of 2L α-Sb, which is in good agreement with the calculated bands. The consistency can also be seen in the spectra taken along 'cut3-5' (Fig. 5b). The zoom-in spectrum along 'cut2' is plotted in

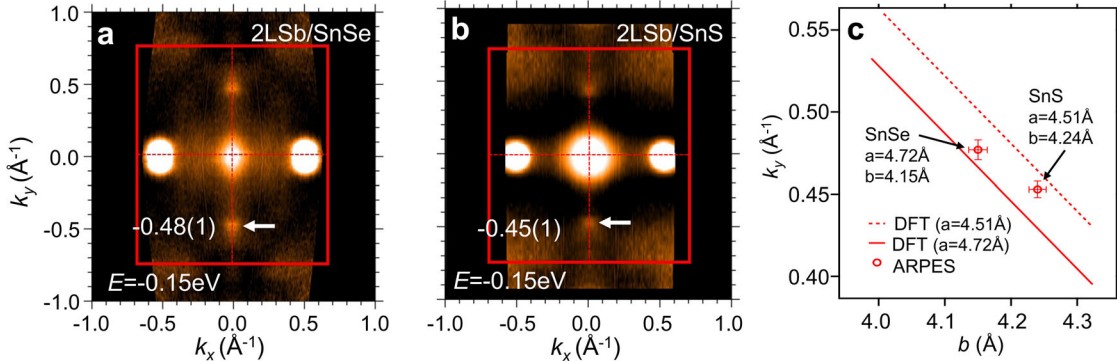

**Fig. 6 | ARPES iso-energy contours of 2L Sb films grown on SnS and SnSe showing the unpinned nature of Dirac states. a** ARPES iso-energy contour at $E = -0.15$ eV taken from the 2L $\alpha$-Sb sample grown on SnSe substrate. The Dirac point of the $p_z$ band is marked by the white arrow. **b** Same as **a**, but from the sample grown on SnS substrate. **c** The location of the Dirac point of the $p_z$ band. The solid line shows the calculated location of the Dirac point for different lattice constant $b$ and a fixed value of $a$, 4.72 Å. The dashed line is same as the solid line but for $a = 4.15$ Å. The open circles indicate the experimental results from the SnS and SnSe samples.

Fig. 5d. Dirac point of the $p_z$ band is at 0.2 eV below the Fermi level. No apparent gap was observed at the Dirac point. The experimental result confirms that the energy gap at the Dirac point of the $p_z$ band is less than the energy resolution of our ARPES instrument (~25 meV).

To demonstrate the unppined nature of the Dirac states, we performed ARPES measurements on 2L $\alpha$-Sb samples grown on a different substrate, SnSe. The epitaxial $\alpha$-Sb films on SnSe and SeS substrates have different lattice parameters due to the lattice strains caused by the substrate effects. According to the STM measurements (see the Supplemental Information), the in-plane lattice parameters of 2L $\alpha$-Sb are $a = 4.72$ Å, $b = 4.18$ Å for the SnSe sample, and $a = 4.51$ Å, $b = 4.27$ Å for the SnS sample. We observed clearly the shift of Dirac points in the SnSe sample compared to the SnS sample. The distance from the Dirac point of the $p_z$ band to the Brillouin zone center is 0.48 Å$^{-1}$ for the SnSe sample and 0.45 Å$^{-1}$ for the SnS sample, as shown in the ARPES iso-energy contours taken from the two samples (Fig. 6a, b). The shift of the Dirac point in momentum space due to the lattice strain is also confirmed by the first-principles calculations as shown in Fig. 6c. Therefore, our ARPES experiments unambiguously establish the existence of the unpinned Dirac states in $\alpha$-Sb.

In summary, our ARPES measurements and first-principles calculations clearly demonstrated that $\alpha$-Sb hosts unpinned Dirac states in both 1L and 2L cases. The Dirac nodes are protected by the spacetime inversion symmetry in the absence of SOC. SOC of the system, on the other hand, induces energy gaps at the Dirac nodes. Surprisingly, we found that the Dirac bands formed by Sb $p_z$ orbitals remain nearly gapless even in the presence of SOC due to the highly suppressed SOC matrix elements. The 2D Dirac nodes at generic $k$-points are unpinned and have highly anisotropic dispersions, which are experimentally confirmed in this study for the first time. The unpinned nature enables versatile ways such as lattice strains to control the locations and the dispersion of the Dirac states. It opens the door to inducing interactions between Dirac states by moving unpinned states closer in momentum space. The Dirac states with tunable properties and controllable couplings are useful for transport and optical applications[29–31]. All of group-Va elements including P, As, Sb and Bi can form phosphorene-like structures[9,10,13–19]. The conduction and valence bands of P and As monolayers are far apart in energy (1.2 eV for P and 0.4 eV for As). Consequently, no Dirac states can form near the Fermi level. On the other hand, the conduction and valence bands of Sb and Bi layers overlap in energy and can generate band crossings near the Fermi level. The size of SOC gaps depends on the effective strength of SOC of the system. The strength of atomic spin-orbit coupling is proportional to the fourth power of the atomic number ($\propto Z^4$). So, Bi has a much stronger SOC coupling compared to Sb, resulting in larger SOC

gaps at the Dirac points of $\alpha$-Bi[8,12]. Therefore, $\alpha$-Sb atomic layers studied in this work provide an ideal platform, compared with other elements in group-Va, for exploring novel properties of unpinned 2D Dirac fermions.

## Methods

### Growth of Sb monolayer and bilayer on SnS and SnSe
$\alpha$-Sb monolayer and bilayer films were grown on the cleaved surface of SnS and SnSe crystals in an MBE-ARPES-STM ultrahigh vacuum (UHV) system. The base pressure was <$2 \times 10^{-10}$ mbar. High-purity Sb was evaporated from a standard Kundsen cell with the flux of 0.3 Å/min. The temperature of the substrate was kept around 105 - 130 °C during the growth. The substrate temperature is critical for growing smooth Sb films in the phosphorene structure.

### Scanning tunneling microscopy measurement
An in-situ Aarhus-150 STM was used to characterize the surface topography and the lattice parameters of the $\alpha$-Sb films. The bias voltage and the tunneling current were set to be 1.5 V and 0.01 nA for the surface topography measurement, 5 mV and 0.15 nA for the zoom-in atom-resolved STM measurement.

### Angle-resolved photoemission spectroscopy measurement
After growth, the samples were in-situ transferred to the ARPES stage under UHV conditions. ARPES measurements were performed at 100 K using a SPECS PHOIBOS-150 hemisphere analyzer. A SPECS UVS-300 helium lamp(He 1$\alpha$ = 21.2 eV) was used as the light source. The energy resolution is around 25 meV at 100 K.

### First-principles calculation
First-principles calculations with Density Function Theory (DFT) were performed by using the Vienna ab Initio Simulation Package (VASP) package[32] The Perdew-Burke-Ernzerhof (PBE)[33] as the exchange-correlation functional was used, and the calculations were performed with plane-wave cut-off energy of 400 eV on the $11 \times 11 \times 1$ Monkhorst Pack k-point mesh. The supercell includes monolayer or bilayer $\alpha$-Sb, and a vacuum layer of thickness about 20 Å, in order to avoid interactions between the neighboring films. The atomic positions were optimized by the conjugate gradient method.

## Data availability
The authors declare that the data supporting the findings of this study are available within the paper and its Supplemental Information files. The data that support the findings of this study are available from the corresponding authors upon reasonable request.

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

## Acknowledgements

G.B. was supported by the US National Science Foundation under grant number NSF DMR-1809160. T.-R.C. was supported by the Young Scholar Fellowship Program from the Ministry of Science and Technology (MOST) in Taiwan, under a MOST grant for the Columbus Program MOST108-2636-M-006-002, National Cheng Kung University, Taiwan, and National Center for Theoretical Sciences, Taiwan. T-.C.C. was supported by the U.S. Department of Energy, Office of Science, Office of Basic Energy Sciences, Division of Materials Science and Engineering, under Grant No. DE-FG02-07ER46383. X.Z. was supported by the fellowship of China Postdoctoral Science Foundation (2021M701590). P.J.K. was supported by National Science Centre, Poland under the project No. 2019/35/B/ST5/03956.

## Author contributions

G.B. and P.J.K. planned the project. Q.L., J.C., and X.Z. synthesized α-Sb film. Q.L., J.C., S.A.B and P.J.K. did the STM measurement. Q.L and X.Z. did the ARPES experiment. Q.L., M.S., K.Y.C., D.T.N, S.A.Y., P.V.S.R., and T.R.C. did the first-principles calculation. B.Q., S.Z., and L.D.Z. provided the SnS and SnSe crystals. Q.L., P.J.K., T.C.C. and G.B. analyzed the data and wrote the paper. All the authors discussed the results and commented on the manuscript.

## Competing interests

The authors declare no competing interests.
