## [Peer Review File · Nature Communications]

Realization of Unpinned Two-Dimensional Dirac States in Antimony Single LayersREVIEWERS' COMMENTS

Reviewer #1 (Remarks to the Author):

In this paper, the authors have reported the observation of so-called unpinned Dirac states, with Dirac points located at low-symmetry k-points in the BZ, in antimony atomic layers with phosphorene structure, confirming an early theoretical prediction. As authors claimed, this is the first experimental observation of such types of Dirac fermions, which is true to my knowledge. The experimental data and results are well presented and convincing, and the paper is well written. Therefore, I recommend for publication, with only one technical point for authors to further clarify:

In general, the global stability of a Dirac point, meaning moving around the BZ without opening a gap, is protected by a spatial symmetry rather than the time-reversal symmetry and inversion symmetry that protect the local stability. For example, for graphene Dirac points at (K, K'), it is the C₃ rotational symmetry. Similarly, the authors may add some discussion on what spatial symmetry (possibly sublattice symmetry since the Dirac points are located away from high-symmetry k-points) is involved in protecting the unpinned Dirac points and how strain is affecting (or not affecting) the symmetry and hence the stability of the Dirac points.

Reviewer #2 (Remarks to the Author):

Authors report discovery of 2D Dirac states in atomic layers of antimony on different substrates. They also demonstrate that momentum location of Dirac points is away from symmetry points of the BZ and can be tuned by strain. This is indeed an important result and deserves publication in Nature Communications as it opens path for tunability of massless surface states. The manuscript is well written, the data and conclusions are sound. I would recommend adding an EDC at $k_x=0$ in Fig. 5 (i. e. for data in panel 5d) to demonstrate absence of a gap at the Dirac point.

Reviewer #3 (Remarks to the Author):

Review-NC

This work highlights a single atomic layer and bi-atomic layers of antimony exhibiting a phosphorene structure. They have carefully grown those films on a SnY (Y=S, Se) substrate and their surfaces are confirmed by a scanning tunneling microscope. The authors have clarified using the first-principles calculation and angle-resolved photoemission that there are some linear band dispersions with crossing points in the vicinity of the Fermi level located at wavenumbers away from high-symmetry points. Their anisotropic features are also confirmed. In particular, the Dirac cones with negligible energy gap are formed in the bi-layer Sb film due to its pz character that is unaffected by the spin-orbit coupling. Finally, they have demonstrated a shift of crossing point in the momentum space by changing the lattice parameter of the films. The experimental datasets shown in the manuscript are reliable and the interpretations are convincing. Therefore, this work certainly deserves publication somewhere.

Despite their main claim on the first finding of unpinned and anisotropic 2D Dirac states. The reviewer thinks that it is not exactly new. It would be said to be new in terms of the fact that the unpinned Dirac states have also been observed in these specific Sb ultra-thin layers. In fact, the non-symmorphic symmetry can cause similar properties. In the prototypical Dirac system, ZrSiS [Nat Commun 7, 11696 (2016)], Dirac cones are formed due to a band folding of the refined BZ due to its non-symmorphic crystal structure. Dirac cones are formed on the two-dimensional square lattice involved in ZrSiS. This is a kind of the first example of the unpinned and anisotropic 2D Dirac states.

Furthermore, despite their statements (p.3, for instance) "Therefore, 2D Dirac states that are unpinned in momentum space are desirable for enabling novel functionalities in Dirac materials", I totally miss the importance of the unpinned and anisotropic 2D Dirac state. There are no

statements or relevant reference papers in this manuscript. Besides, although the authors claim "Therefore, the Dirac states in group-Va few-layers are distinct from those in graphene and thus, offer new insights into the Dirac fermion physics at low dimensions. (p.3)", the reviewer wonders what kind of new insights one could expect. There are no concrete examples raised in the manuscript.

Having considered the above-mentioned points, I conclude that this paper does not meet the high standard of Nature Communications.

Summary of Changes:

1. Add a statement about stability of Dirac points in antimonene as suggested by Reviewer 1.
2. Add an EDC curve to Figure 5 as suggested by Reviewer2.
3. Add a statement about ZrSiS and the reference (Nat Commun 7, 11696 (2016)) pointed out by the Reviewer 3. “

Reviewer #1:

In this paper, the authors have reported the observation of so-called unpinned Dirac states, with Dirac points located at low-symmetry k-points in the BZ, in antimony atomic layers with phosphorene structure, confirming an early theoretical prediction. As authors claimed, this is the first experimental observation of such types of Dirac fermions, which is true to my knowledge. The experimental data and results are well presented and convincing, and the paper is well written. Therefore, I recommend for publication, with only one technical point for authors to further clarify:

Authors: We thank the reviewer for the insightful comments and the recommendation.

In general, the global stability of a Dirac point, meaning moving around the BZ without opening a gap, is protected by a spatial symmetry rather than the time-reversal symmetry and inversion symmetry that protect the local stability. For example, for graphene Dirac points at (K, K'), it is the C3 rotational symmetry. Similarly, the authors may add some discussion on what spatial symmetry (possibly sublattice symmetry since the Dirac points are located away from high-symmetry k-points) is involved in protecting the unpinned Dirac points and how strain is affecting (or not affecting) the symmetry and hence the stability of the Dirac points.

Authors: We agree with the reviewer that the discussion on the stability of Dirac points is very important. We have added the following statements to the text:

“The Dirac states in those 2D systems are exclusively pinned at high-symmetry points of the Brillouin zone, such as K(K') of graphene (due to the C3 rotational symmetry). The Dirac states also feature isotropic low-energy dispersion due to the local rotational symmetry of the crystal lattice.”

“The Dirac nodes in α -Sb are protected by the spacetime inversion symmetry in the absence of SOC. SOC of the system, on the other hand, induces energy gaps at the Dirac nodes. Surprisingly, we found that the Dirac bands formed by Sb p_z orbitals remain nearly gapless even in the presence of SOC due to the highly suppressed SOC matrix elements. The 2D Dirac nodes at generic k-points are unpinned and have highly anisotropic dispersions, which are experimentally confirmed in this study for the first time. The unpinned nature enables versatile ways such as lattice strains to control the locations and the dispersion of the Dirac states.”

Reviewer #2:

Authors report discovery of 2D Dirac states in atomic layers of antimony on different substrates. They also demonstrate that momentum location of Dirac points is away from symmetry points of the BZ and can be tuned by strain. This is indeed an important result and deserves publication in

Nature Communications as it opens path for tunability of massless surface states. The manuscript is well written, the data and conclusions are sound. I would recommend adding an EDC at $k_x=0$ in Fig. 5 (i. e. for data in panel 5d) to demonstrate absence of a gap at the Dirac point.

Authors: We thank the reviewer for the insightful comments and the recommendation. We have added an EDC to Fig. 5 as suggested by the reviewer.

Reviewer #3:

This work highlights a single atomic layer and bi-atomic layers of antimony exhibiting a phosphorene structure. They have carefully grown those films on a SnY (Y=S, Se) substrate and their surfaces are confirmed by a scanning tunneling microscope. The authors have clarified using the first-principles calculation and angle-resolved photoemission that there are some linear band dispersions with crossing points in the vicinity of the Fermi level located at wavenumbers away from high-symmetry points. Their anisotropic features are also confirmed. In particular, the Dirac cones with negligible energy gap are formed in the bi-layer Sb film due to its pz character that is unaffected by the spin-orbit coupling. Finally, they have demonstrated a shift of crossing point in the momentum space by changing the lattice parameter of the films. The experimental datasets shown in the manuscript are reliable and the interpretations are convincing. Therefore, this work certainly deserves publication somewhere.

Authors: We thank the reviewer for the careful review and insightful comments. We have revised the manuscript according to the reviewers' comments.

Despite their main claim on the first finding of unpinned and anisotropic 2D Dirac states. The reviewer thinks that it is not exactly new. It would be said to be new in terms of the fact that the unpinned Dirac states have also been observed in these specific Sb ultra-thin layers. In fact, the non-symmorphic symmetry can cause similar properties. In the prototypical Dirac system, ZrSiS [Nat Commun 7, 11696 (2016)], Dirac cones are formed due to a band folding of the refined BZ due to its non-symmorphic crystal structure. Dirac cones are formed on the two-dimensional square lattice involved in ZrSiS. This is a kind of the first example of the unpinned and anisotropic 2D Dirac states.

Authors: We thank the reviewer for pointing out the important reference, and we agree with the reviewer that a similar band dispersion was observed in bulk crystal ZrSiS. Here we respectfully point out the critical differences between the paper on ZrSiS and our work.

The novelty of unpinned Dirac states we observed in antimonene is as follows: (1) the system is purely 2D, as in 1L and 2L Sb layers; (2) the band dispersion is nearly gapless as the SOC gap is highly suppressed in 2L Sb; (3) the unpinned nature of Dirac states (shifting in momentum space) is demonstrated in our experiments. By contrast, the previous work on ZrSiS focused on the bulk crystal, in which a weak dispersion in the k_z direction was observed. A prominent SOC gap was found in the Dirac bands of ZrSiS. No unpinned feature was demonstrated in that paper. Therefore, our work presents the first conclusive evidence for 2D unpinned Dirac states, a new type of Dirac materials beyond graphene.

We have added the paper (Nat. Commun. 7, 11696 (2016)) to the references and included the following statement,

“The gapped band dispersion is similar to the quasi-2D Dirac states observed in bulk crystal ZrSiS [26].”

Furthermore, despite their statements (p.3, for instance) “Therefore, 2D Dirac states that are unpinned in momentum space are desirable for enabling novel functionalities in Dirac materials”, I totally miss the importance of the unpinned and anisotropic 2D Dirac state. There are no statements or relevant reference papers in this manuscript. Besides, although the authors claim “Therefore, the Dirac states in group-Va few-layers are distinct from those in graphene and thus, offer new insights into the Dirac fermion physics at low dimensions. (p.3)”, the reviewer wonders what kind of new insights one could expect. There are no concrete examples raised in the manuscript. Having considered the above-mentioned points, I conclude that this paper does not meet the high standard of Nature Communications.

Authors: We thank the reviewer for the comment. The Dirac states in the known 2D systems are exclusively pinned at high-symmetry points of the Brillouin zone. This feature imposes constraints on applications of the massless Dirac states. For example, the two Dirac cones in graphene are pinned at opposite corners of the Brillouin zone. It is difficult to make two valleys effectively coupled in monolayer graphene. Therefore, 2D Dirac states that are unpinned in momentum space are desirable for enabling novel functionalities in Dirac materials.

In our work, we observed the Dirac states are located at generic momentum points in Sb layers. The location of Dirac points can be shifted by lattice strain, as confirmed by our ARPES experiments. This unpinned nature enables versatile ways such as lattice strains to control the locations and the dispersion of the Dirac states. It opens the door to inducing interactions between Dirac states by moving unpinned states closer in momentum space. The Dirac states with tunable properties and controllable couplings are useful for transport and optical applications. Reviewer 2 points out that “this is indeed an important result, ..., as it opens path for tunability of massless surface states.”

We have added the following statements to the text:

“The Dirac states in the known 2D systems are exclusively pinned at high-symmetry points of the Brillouin zone. This feature imposes constraints on applications of the massless Dirac states. For example, the two Dirac cones in graphene are pinned at opposite corners of the Brillouin zone. It is difficult to make two valleys effectively coupled in monolayer graphene. Therefore, 2D Dirac states that are unpinned in momentum space are desirable for enabling novel functionalities in Dirac materials.”

“In our work, we observed the Dirac states are located at generic momentum points in Sb layers. The location of Dirac points can be shifted by lattice strain, as confirmed by our ARPES experiments. This unpinned nature enables versatile ways such as lattice strains to control the

locations and the dispersion of the Dirac states. It opens the door to inducing interactions between Dirac states by moving unpinned states closer in momentum space. The Dirac states with tunable properties and controllable couplings are useful for transport and optical applications.”